# (Sb_0.5_Li_0.5_)TiO_3_-Doping Effect and Sintering Condition Tailoring in BaTiO_3_-Based Ceramics

**DOI:** 10.3390/ma17092085

**Published:** 2024-04-29

**Authors:** Juanwen Yan, Bijun Fang, Shuai Zhang, Xiaolong Lu, Jianning Ding

**Affiliations:** 1School of Materials Science and Engineering, Jiangsu Collaborative Innovation Center of Photovoltaic Science and Engineering, Jiangsu Province Cultivation Base for State Key Laboratory of Photovoltaic Science and Technology, National Experimental Demonstration Center for Materials Science and Engineering, Changzhou University, Changzhou 213164, China; yanjuanwen123@163.com (J.Y.); shuaizhang@cczu.edu.cn (S.Z.); xllu@cczu.edu.cn (X.L.); 2School of Mechanical Engineering, Yangzhou University, Yangzhou 225127, China

**Keywords:** (Sb_0.5_Li_0.5_)TiO_3_ doping, sintering condition, relaxation characteristic, electrical properties, thermal stability

## Abstract

(1-x)(Ba_0.75_Sr_0.1_Bi_0.1_)(Ti_0.9_Zr_0.1_)O_3_-x(Sb_0.5_Li_0.5_)TiO_3_ (abbreviated as BSBiTZ-xSLT, x = 0.025, 0.05, 0.075, 0.1) ceramics were prepared via a conventional solid-state sintering method under different sintering temperatures. All BSBiTZ-xSLT ceramics have predominantly perovskite phase structures with the coexistence of tetragonal, rhombohedral and orthogonal phases, and present mainly spherical-like shaped grains relating to a liquid-phase sintering mechanism due to adding SLT and Bi_2_O_3_. By adjusting the sintering temperature, all compositions obtain the highest relative density and present densified micro-morphology, and doping SLT tends to promote the growth of grain size and the grain size distribution becomes nonuniform gradually. Due to the addition of heterovalent ions and SLT, typical relaxor ferroelectric characteristic is realized, dielectric performance stability is broadened to ~120 °C with variation less than 10%, and very long and slim hysteresis loops are obtained, which is especially beneficial for energy storage application. All samples show extremely fast discharge performance where the discharge time t_0.9_ (time for 90% discharge energy density) is less than 160 ns and the largest discharge current occurs at around 30 ns. The 1155 °C sintered BSBiTZ-0.025SLT ceramics exhibit rather large energy storage density, very high energy storage efficiency and excellent pulse charge–discharge performance, providing the possibility to develop novel BT-based dielectric ceramics for pulse energy storage applications.

## 1. Introduction

In recent years, global energy poverty has been increasing. It is well known that renewable energy can not only directly alleviate global energy poverty, but also have a significant inhibitory effect on energy poverty by improving energy efficiency [1]. Then, the development, storage and utilization of clean and renewable energy sources are becoming increasingly significant [2]. Among many energy storage devices, lead-free dielectric ceramics have plenty of outstanding advantages: faster charge–discharge rate, higher breakdown field strength, less expensive cost, friendly to the environment and health and a wider range of applications [3,4,5,6]. Therefore, exploring suitable lead-free dielectric ceramics to store energy provides a key to solving the problem.

As a representative of lead-free ferroelectric materials with perovskite structure, the BaTiO_3_ (BT) ceramic has a dielectric constant of about 2000 at room temperature and more than 8000 at the Curie temperature (T_C_) of about 120 °C, attracting extensive attention in the fields of ceramic capacitors, positive temperature coefficient (PTC) thermistors and piezoelectric devices [7]. However, the pure BT ceramic has two main shortcomings. Firstly, the pure BT ceramic possesses long-range ordered ferroelectric domains at room temperature, which reaches nonlinear saturation polarization at a low electric field and has high remnant polarization, leading to large energy loss and low energy storage efficiency. Secondly, the pure BT ceramic has weak dielectric temperature stability due to the ferroelectric-to-paraelectric phase transition around the T_C_ temperature [8]. As a consequence, it is necessary to modify the dielectric and energy storage properties of pure BT to meet the requirements of miniaturization, integration and stability of energy storage devices [7,8].

According to the formula of energy storage density [4,6]:(1)$$W_{rec}=\int_{P_{r}}^{P_{max}}EdP$$
where *W_rec_*, *P_max_*, *P_r_* and *E* represent recoverable energy storage density, maximum polarization, remnant polarization and external electric field, respectively, expanding the polarization difference *ΔP* (*ΔP* = *P_max_* − *P_r_*) and increasing the breakdown field strength (BDS) of ceramics can improve energy storage performance [9].

In previous studies, matrix solid solutionizing and ionic doping are the two most commonly used methods to tailor normal ferroelectric ceramics into relaxor ferroelectric ceramics. Due to doping ions with unequal ionic radius or unequal valence state, the structure of the oxygen octahedron is distorted and the long-range ordered ferroelectric domains are broken into polar nano-regions (PNRs), which can increase the response to an external electric field, reduce P_r_ and broaden the dielectric constant peak [10,11]. Heterovalent ion doping also induces defect dipoles, which reduces grain size by inhibiting the movement of carriers, decreases leakage current and dielectric loss and increases BDS. In addition, point defects and secondary phases are attracted to grain boundaries and form space charges, which will generate space charge polarization and enhance P_max_ [12]. For example, Yan et al. found that BDS increased from 39.2 kV/cm in (Ba_0.9_Li_0.1_)TiO_3_ to 76.5 kV/cm in (Ba_0.6_Li_0.4_)TiO_3_, and the W_rec_ value increased from 0.089 J/cm^3^ to 0.293 J/cm^3^ [12].

Compared with doping isolated ions, the effect of ion pairs on chemical compressive stress is more remarkable. In previous studies, co-doped Li^+^ and Al^3+^ at the A site of BaTiO_3_ or Pb(Zr, Ti)O_3_ (PZT) ceramics tend to form Li^+^-Al^3+^ ion pairs along the [001] direction, which will introduce large uniaxial chemical pressure due to local lattice distortion and have a great influence on the phase transformation behavior and piezoelectric properties of the BT ferroelectric ceramics [13]. Alkathy et al.’s study determined that the *W_rec_* value of Sr_0.92_(Nd, Li)_0.08_TiO_3_ was 9 times higher than SrTiO_3_, from 0.11 J/cm^3^ to 0.952 J/cm^3^, and the energy storage efficiency *η* ($\eta =W_{rec}/W_{t}\times 100\%$, *W_t_* is total energy storage density) increased from 88.71% to 95.98%, accompanied by the grain size decreased from 10.218 μm to 3.46 μm [14]. Alkathy et al. prepared the (Ba_0.60_Sr_0.40_)_0.92_(Bi, Li)_0.08_TiO_3_ ceramics by the conventional solid-phase reaction method, which possessed an ultra-high dielectric constant of more than 10^5^, and the *W_rec_* and *η* values were 0.3856 J/cm^3^ and 93.26% at 100 kV/cm, respectively [15]. The high output power (9.63 μW/cm^2^) and energy density (0.93 μJ/cm^3^) were generated in the 0.98(Na_0.5_K_0.5_)NbO_3_-0.02LiSbO_3_ ceramics sintered at 1100 °C reported by Kim et al. and the tetragonality was gradually strengthened, along with increasing the LiSbO_3_ content [16]. Rubenis et al. found that the Sb^3+^ doping reduced the grain size and made the microstructure more homogeneous [17]. Therefore, adding heterovalent ions Sb^3+^ and Li^+^ and second component SLT into the BT-based ceramics was undertaken in this work to realize the relaxation characteristic, where Li improves relaxation performance due to inequality of electric valence and ionic radius, Sb balances the electrovalence caused by Li doping, strengthens the tetragonal phase and increases microscopic morphology homogeneity, and SLT not only improves the relaxation properties but also reduces the sintering temperature.

It is well known that the preparation method and ceramic processing are closely related to the properties and structure of ceramics, among which the research on the influence of sintering temperature is the basis of all studies [18,19,20]. Li et al. obtained small grain size, narrow grain size distribution, uniform morphology and high-density Pb(Zr_0.5_Ti_0.5_)O_3_-based ceramics due to low sintering temperatures [21]. He et al. found nanoscale grain size in the [(Ba_0.85_Ca_0.15_)_0.995_Nd_0.005_](Ti_0.9_Hf_0.1_)O_3_ ceramics prepared via hydrothermal method exhibited an increasing trend with the increase in sintering temperature and holding time, which significantly affected the dielectric constant and T_C_ temperature [22]. The T_C_ temperature of La-doped BT prepared by Kuwabara et al. ascended to 3.5 °C with an elevating sintering temperature from 1300 °C to 1450 °C, accompanied by ramping up of the volume resistivity and the total resistivity at room temperature [19]. Therefore, this study aims to obtain high density, small grain size and uniform microscopic morphology that can be obtained by adjusting the sintering temperature. At the same time, the T_C_ temperature can be adjusted below the room temperature and the resistance of ceramics can be improved, combined with composition designing.

In order to develop novel green and environment-friendly lead-free dielectric ceramics for pulse energy storage applications, the BT-based ceramics were studied by adding heterovalent ions Sb^3+^ and Li^+^ and second component SLT. The effects of sintering temperature and the (Sb_0.5_Li_0.5_)TiO_3_ addition amount on the structure, morphology and electrical properties of (Ba_0.75_Sr_0.1_Bi_0.1_)(Ti_0.9_Zr_0.1_)O_3_ were researched systematically, and the basic mechanism of enhanced thermal stability was revealed. Eventually, in this work, the densified sintering with a relative density exceeding 95% is realized in the (1-x)(Ba_0.75_Sr_0.1_Bi_0.1_)(Ti_0.9_Zr_0.1_)O_3_-x(Sb_0.5_Li_0.5_)TiO_3_ system, and the 1155 °C sintered 0.975(Ba_0.75_Sr_0.1_Bi_0.1_)(Ti_0.9_Zr_0.1_)O_3_-0.025(Sb_0.5_Li_0.5_)TiO_3_ ceramics have excellent energy storage performance, which shows good temperature stability of the dielectric constant until ~120 °C with a variation of less than 10%, excellent energy storage performance with *W_rec_* = 65.59 mJ/cm^3^ and *η* = 97.02% under 50 kV/cm, and the discharge current (*I_d_*) reaches the maximum value at around 30 ns, providing remarkable application value in the fields of pulse power supply and electric vehicles [23].

## 2. Experimental Procedure

Bulk (1-x)(Ba_0.75_Sr_0.1_Bi_0.1_)(Ti_0.9_Zr_0.1_)O_3_-x(Sb_0.5_Li_0.5_)TiO_3_ (abbreviated as BSBiTZ-xSLT, x = 0.025, 0.05, 0.075, 0.1) ceramics were prepared by conventional solid-state reaction method. The raw reagents, BaCO_3_ (99%, Sinopharm Chemical Reagent Co., Ltd., Shanghai, China), SrCO_3_ (99%, Sinopharm Chemical Reagent Co., Ltd., Shanghai, China), Bi_2_O_3_ (99%, Sinopharm Chemical Reagent Co., Ltd., Shanghai, China), TiO_2_ (98.5%, Sinopharm Chemical Reagent Co., Ltd., Shanghai, China), ZrO_2_ (99%, Sinopharm Chemical Reagent Co., Ltd., Shanghai, China), Sb_2_O_3_ (99%, Shanghai Shisihewei Chemical Co., Ltd., Shanghai, China) and Li_2_CO_3_ (98%, Shanghai Shanhai Gongxue Group Experiment No.2 Factory, Shanghai, China), were stoichiometrically weighed after full drying, fully ground and mixed and passed through a 100-mesh sieve. The sieved powder was calcined in a muffle furnace whose temperature was increased to 500 °C for 167 min, then raised to 925 °C at 5 °C/min and, finally, kept at 925 °C for 3 h. Polyvinyl alcohol (PVA) solution was mixed with the calcined powder for granulation, and the granulated powder was passed through an 80-mesh sieve. The granulated powder was pressed into green pellets with a diameter of 10 mm and thickness of about 1 mm via cold pressing under a pressure of 350 MPa and a holding time of 1 min. Subsequently, the pressed discs were placed in a muffle furnace, increased to 550 °C for 200 min and incubated for 2 h to remove the PVA. Then, using ZrO_2_ as the covering powder to sinter the decarburized pressed discs at different temperatures with a holding time of 2 h and the sintered ceramics are shown in Figure 1. Several sintering temperatures were treated using 15 °C as an interval depending on composition, and the sintering temperature range and optimized sintering temperature were determined based on density measurement and performance characterization.

The crystal structure of the BSBiTZ-xSLT ceramics was determined by X-ray diffraction measurement (XRD, Rigaku D/max-2500/PC, Rigaku Corp., Tokyo, Japan). The peak splitting and accurate phase structure were analyzed by the WinPLOTR-2006 software. The theoretical density and the composition-induced phase transformation were acquired by the MDI Jade 6.5 software and Raman spectroscopy (LabRAM HR Evolution, Horiba Scientific, Ltd., Kyoto, Japan) using a laser with a wavelength of 633 nm in the range of 100~1000 cm^−1^. Scanning electron microscopy (SEM, JSM-IT100, JEOL Ltd., Tokyo, Japan) and Nano Measurer 1.2 software were used to obtain the microstructure and grain size distribution, respectively. Dielectric properties were tested from room temperature to 180 °C at 100 Hz to 2 MHz using a Partulab HDMS-1000 measurement system (Partulab Technology Co. Ltd., Wuhan, China) combined with a Microtest Precision LCR Meter 6630-10 (Microtest Corp., Taiwan, China). The ferroelectric properties were tested at 10 Hz using a ferroelectric analyzer (Radiant Technologies Inc., Albuquerque, NM, USA). The resistance–inductance–capacitance test system (Shanghai Tongguo Intelligent Technology Co., Ltd., Shanghai, China) was used to test out the pulse charge–discharge performance.

## 3. Results and Discussion

### 3.1. Structure Change and Density Analysis

In order to understand the relationship between composition, sintering temperature and phase structure, the XRD patterns of BSBiTZ-xSLT (x = 0.025, 0.05, 0.075, 0.1) at different sintering temperatures are shown in Figure 2a and Appendix A. It can be seen that all ceramics show the predominant perovskite phase structure with a small amount of secondary phase BaTi_2_O_5_, which may be attributed to the volatilization of volatile Bi^3+^ and Li^+^ ions during high-temperature sintering. Due to the decreased solubility of Bi^3+^ ion and adding many cations in the BT-based ceramics, the secondary phase BaTi_2_O_5_ appears [24]. As the sintering temperature increases, the Bi^3+^ and Li^+^ ions will volatilize even more, resulting in the increase in XRD intensity of BaTi_2_O_5_ in corresponding ceramics accordingly, which will enhance the conductivity and make the samples break down easily [25,26]. For the BSBiTZ-0.025SLT system, the splitting of {110} crystal plane becomes apparent when the ceramics are sintered above 1170 °C, presenting tetragonal and orthogonal crystal system characteristics. With the increase in the SLT second component amount, the {110} splitting becomes weak and disappears completely in the BSBiTZ-0.1SLT system even at the highest sintering temperature undertaken, showing a signature of composition-induced phase transformation.

Figure 2b,c are the amplified diffraction patterns around the {111} and {200} crystal planes. When the sintering temperature is 1185 °C, the {111} and {200} diffraction reflections are symmetric and have no splitting, indicating that the phase structure approaches the pseudo-cubic phase. When the sintering temperature is less than 1185 °C, {111} splits into (111) and $\left(\bar{1}11\right)$ peaks, indicating that the ceramics contain the rhombohedral phase [27], and {200} presents doublet splitting with higher intensity for the lower 2θ peak or triplet-like splitting, suggesting that the structure may contain the tetragonal and orthogonal phases [28]. With the increase in sintering temperature, the broadening and splitting of crystal planes gradually become weak or disappear, indicating that the structure changes to a pseudo-cubic phase. Moreover, the {111} and {200} reflections tend to shift to a lower 2θ angle, correlating with the lattice expansion caused by the growth of crystal grains with elevating sintering temperature, and being consistent with the literature report [29].

XRD patterns of the BSBiTZ-xSLT (x = 0.025, 0.05, 0.075, 0.1) ceramics sintered at 1155 °C, 1155 °C, 1155 °C and 1170 °C are shown in Figure 3a. All BSBiTZ-xSLT ceramics have a predominant perovskite structure with slight impurity BaTi_2_O_5_. Figure 3b,c show the amplified {111} and {200} crystal planes, in which {200} has triplet splitting characteristics and is deconvoluted into three peaks by the WinPLOTR software fitting. With the increase in the SLT amount, the broadening or splitting of {111} gradually disappears, and the relative intensity of the {200} splitting peaks changes accordingly, confirming the occurrence of a composition-induced phase transition. Therefore, BSBiTZ-xSLT is located at the morphotropic phase boundary region, where rhombohedral, tetragonal and orthogonal phases co-exist, which will improve the relaxor characteristic [28]. In addition, the diffraction peaks shift to a larger 2θ angle first and then move to a smaller 2θ angle, attributing to two reasons. The ionic radius of doped ions at the A site of the perovskite structure is 0.76 Å (6-coordination number, CN) for Sb^3+^, and 0.92 Å (8-CN) for Li^+^, which is smaller than the original ionic radius at the A site of the matrix, at 1.61 Å (12-CN) for Ba^2+^, 1.44 Å (12-CN) for Sr^2+^ and 1.38 Å (12-CN) for Bi^3+^, resulting in shrinkage of a unit cell and a decrease in lattice volume; then, the diffraction peak first moves to a higher 2θ angle. When the SLT doping amount reaches x = 0.1, the grain size increases, and lattice distortion or stain changes, leading the diffraction peak shift to a smaller 2θ angle [30].

The multi-phase co-existing structure in the BSBiTZ-xSLT ceramics can be verified further by Raman spectroscopy [31,32]. Figure 4 shows Raman spectra of the BSBiTZ-xSLT (x = 0.025, 0.05, 0.075, 0.1) ceramics sintered at 1155 °C, 1155 °C, 1155 °C and 1170 °C, respectively, from 100 cm^−1^ to 1000 cm^−1^. Six main vibrational modes are found in all samples: ν_r_ at 120 cm^−1^ and ν_o_ at 187 cm^−1^ are related to A site. The vibration of the B-O bonds generates Raman modes at 261 cm^−1^ and 303 cm^−1^ wavenumbers, proving that the Sb^3+^ and Li^+^ ions are dissolved in the A site. The densely dispersed vibrational modes corresponding to the stretching and bending of the A site and B-O bonds show that the substitution of the A and B positions leads to an increase in ion disorder with increasing x value. The ν_r_ peak and ν_o_ peak indicate the presence of the rhombohedral phase and orthogonal phase in the structure [3,33]. In addition, the gradual flattening of the terminal valleys of the A site and B-O bonds is also related to internal stress and lattice defects. The sharp peak within 450 cm^−1^–650 cm^−1^ is related to the vibration of the [BO_6_] octahedron. The 516 cm^−1^ peak moves to the lower wavenumber direction and tends to become broadening when the SLT amount increases from 0.025 to 0.075. Such phenomena reflect the structural evolution trend of enhancement of pseudo-cubic phase isotropy and inherent static disorder due to the doping of Zr^4+^ and Ti^4+^, which improves the dielectric performance [4]. When the x value reaches 0.1, the 516 cm^−1^ peak suddenly becomes stronger and narrower as compared with the BSBiTZ-0.05SLT ceramics, attributing to the appearance of pseudo-cubic phase accompanied by the change of domain configuration and relaxor characteristic as discussed below, where the ferroelectric-to-paraelectric phase transition is affected greatly by the mechanical and electrical strain caused by the high concentration point defects in the BT-based ceramics due to the heterovalent ions doping and adding SLT [34]. When the wavenumber is greater than 700 cm^−1^, it is the superposition of the A_1_+E longitudinal optical modes [32] and the ν_t_ mode at 729 cm^−1^ is the characteristic peak of the tetragonal phase [3,32,33].

Based on WinPLOTR fitting and Jade refinement, the theoretical density is calculated as shown in Table 1, combined with bulk density achieved by the water immersion method, for the BSBiTZ-xSLT (x = 0.025, 0.05, 0.075, 0.1) ceramics prepared at different sintering temperatures. The relative density of all compositions increases first and reaches a peak value at slightly different sintering temperatures, which are 1155 °C, 1155 °C, 1155 °C and 1170 °C for x = 0.025, 0.05, 0.075 and 0.1 compositions, respectively, and then decreases with the increase in sintering temperature, showing a narrow sintering temperature range. Such change characteristic is related to the accelerated diffusion rate of carriers with increasing sintering temperature, which promotes the mutual fusion between particles and increases the density of ceramics [35]. If the sintering temperature is too high, over-sintering will occur, grains will grow faster and abnormal grain growth will appear, which will disrupt the combination of the original particles, and some cations will evaporate excessively, leading to the increase in pores or even the forming of cracks in the ceramics. The peak relative density of all compositions exceeds 96%, increasing breakdown endurance and being suitable for industry application [25]. Additionally, the peak relative density tends to increase with elevating the SLT amount, showing the promotion of sintering densification characteristic of the Bi, Sb and Li ions.

### 3.2. SEM Morphology and Grain Size Distribution

SEM photos and corresponding grain size distribution statistical curves of the BSBiTZ-0.025SLT ceramics measured by a linear intersection method using the Nano Measure software sintered at 1125 °C, 1140 °C, 1155 °C, 1170 °C and 1185 °C are shown in Figure 5. Most grains show a nearly spherical shape, demonstrating the important role of the liquid-phase sintering mechanism in densification due to the addition of SLT and Bi elements [3,21]. With the increase in sintering temperature, gas pores decrease first, and reach a minimum at a sintering temperature of 1155 °C, whose sample has the largest relative density of 96.60%, and then increase, coinciding well with the decrease in relative density. The average grain size increases gradually from 0.88 μm at 1125 °C to 1.52 μm at 1185 °C, and grain size distribution departs from the normal distribution accompanied by obvious abnormal grain growth above sintering temperature 1155 °C, since excessively higher sintering temperature breaks the equilibrium between grain growth and pore extrusion, and promotes anomalous grain growth [29].

Point defects caused by doping with heterovalent ions exert additional influence as shown below by defect chemical reaction equations in the matrix [21]:(2)$${Li}_{2}CO_{3}\overset{BT}{\to }2{Li}_{Ba}^{\prime }+V_{O}^{\cdot \cdot }+CO_{2}↑+O_{o}^{\times }$$
(3)$${Sb}_{2}O_{3}\overset{BT}{\to }2{Sb}_{Ba}^{\cdot }+V_{Ba}^{\prime \prime }+3O_{o}^{\times }$$
(4)$${Bi}_{2}O_{3}\overset{BT}{\to }2{Bi}_{Ba}^{\cdot }+V_{Ba}^{\prime \prime }+3O_{o}^{\times }$$
(5)$${Ba}_{Ba}^{\times }+O_{o}^{\times }\to Ba↑+\frac{1}{2}O_{2}↑+V_{Ba}^{\prime \prime }+V_{O}^{\cdot \cdot }$$

Such nonequivalent ions doping generates a large number of point defects, which will overcome the binding barrier and disperse out of the equilibrium position once obtaining high enough external energy.

According to the diffusion coefficient formula [35]:(6)$$D=D_{0}\exp\;(-\frac{Q}{RT})$$

Charged carriers diffuse easier at higher sintering temperatures, producing larger grain sizes and promoting abnormal grain growth [35]. Hence, the rise of sintering temperature will accelerate the migration of carriers and promote grain growth. The growth of grain size is conducive to acquiring dense microstructure, which will increase breakdown field strength. However, when the sintering temperature is too high, abnormal grain growth appears, excessive sublimation of Bi, Sb and Li elements occurs, grain size increases are too large and grain boundary density becomes low, which will reduce mechanical performance and breakdown field strength of the ceramics.

Figure 6 shows SEM images and grain size distribution of the BSBiTZ-xSLT compositions, where x = 0.05, 0.075 and 0.1 samples are sintered at 1155 °C, 1155 °C and 1170 °C, respectively. Combined with Figure 5(c1,c2), all BSBiTZ-xSLT ceramics with the largest relative density in respective compositions present densified micro-morphology and mainly spherical-like shaped grains due to the low melting point of Bi_2_O_3_ [21], inducing liquid-phase sintering and promoting grain growth of Li_2_CO_3_ and Sb_2_O_3_ [13,17]. The BSBiTZ-0.025SLT and BSBiTZ-0.05SLT ceramics exhibit rather homogenous grain morphology, whereas BSBiTZ-0.075SLT has a large number of grains with small grain size, and BSBiTZ-0.1SLT presents bimodal grain size distribution and abnormal grain growth. Grain size shows a negative correlation with grain boundary density, which is affected by liquid-phase sintering and point defects caused by heterovalent ion doping. Heterovalent ion doping forms defect dipoles, which not only influence charged carriers’ diffusion and inhibit grain growth [15,25], but also reduce leakage current, increase BDS and enhance P_max_ owing to generation of space charge [15]. Comparatively speaking, sintering temperature has a greater influence on grain growth than the SLT doping amount in the BSBiTZ-xSLT system, and BSBiTZ-0.025SLT is an optimal solid solution from a microscopic morphology point of view.

### 3.3. Dielectric Properties

Sintering temperature and composition affect the grain size, crystallinity and porosity of ceramics, presenting a remarkable impact on the dielectric properties [21]. Figure 7a shows the dielectric performance–temperature relationship of the BSBiTZ-0.025SLT ceramics sintered at different temperatures measured at 10 kHz. Due to the addition of heterovalent ions and SLT, the dielectric peak is compressed, T_m_ (temperature of maximum dielectric constant) is moved to below room temperature, and the dielectric constant increases first from 682 at 1125 °C to 705 at 1155 °C and then drops to 550 at 1185 °C with the increase in sintering temperature. Generally speaking, dielectric loss is rather low, all less than 0.04, and the incomplete broad loss tangent peak around ambient temperature for some samples also reveals that the ferroelectric phase transition occurs below room temperature. More importantly, the temperature stability of the dielectric constant is expanded to ~120 °C and the variation is less than 10%, which is especially conducive to energy storage applications. The dielectric constant reaches the maximum value at a sintering temperature of 1155 °C, which can be assigned to the fine grain size and morphology uniformity, excellent crystallinity, less porosity and highest density (Figure 5 and Table 1). When the sintering temperature exceeds 1170 °C, porosity increases again and abnormal grain growth appears, leading to a decrease in the dielectric constant.

The influence of composition on temperature-dependent dielectric performance of the BSBiTZ-xSLT (x = 0.025, 0.05, 0.075, 0.1) ceramics at 10 kHz is shown in Figure 7b. All samples present excellent dielectric performance thermal stability, and the dielectric constant rises first and then reduces with elevating the SLT content. Besides the change of density, Li^+^, Bi^3+^, etc., heterovalent ion doping generates defect dipoles, and grain boundary density increases due to the decrease in grain size, attracting more defect dipoles. These features change with composition and generate space charge polarization; as a consequence, BSBiTZ-0.05SLT exhibits the highest dielectric constant. BSBiTZ-0.1SLT and BSBiTZ-0.075SLT display the evident influence of grain size and nonuniformity, whose dielectric constant is rather small, whereas their thermal stability becomes very excellent.

Temperature-dependent dielectric performance at several frequencies of the BSBiTZ-xSLT (x = 0.025, 0.05, 0.075, 0.1) ceramics at various sintering temperatures are shown in Figure 8 and Appendix A. All ceramics present obvious dielectric frequency dispersion and their dielectric performance shows high dependency on frequency change, especially in the loss tangent–temperature curves, indicating that the relaxation characteristic is achieved in the BT-based ceramics due to the heterovalent ions doping and addition of SLT, generating space charge and responding different polarization to frequency [27]. The degree of relaxation diffusion and frequency dispersion becomes more apparent with a diminishing x value, but all samples still have excellent frequency stability above 100 °C. Due to heterovalent ions doping and adding SLT, the T_m_ temperature shifts to below room temperature, the dielectric constant peak is broadened greatly and a wide dielectric plateau appears, showing good temperature stability [36]. Such results indicate that the long-range ordered macroscopic ferroelectric domains in the original pure BT ceramics are broken, which gradually transform into short-range polar nano-regions (PNRs) in the BSBiTZ-xSLT ceramics, causing the increase in frequency dispersion and relaxation characteristic [37] and the decrease in the dielectric constant [27].

### 3.4. Ferroelectric and Pulse Energy Storage Performance

Figure 9 and Appendix A show unipolar polarization–electric field (P-E) hysteresis loops of the BSBiTZ-xSLT (x = 0.025, 0.05, 0.075, 0.1) ceramics sintered at different temperatures measured at room temperature, 10 Hz and 10–50 kV/cm. As a result of heterovalent ions doping and adding SLT, the long-range ordered ferroelectric domains are destroyed and PNRs are formed due to the different valence states and ionic radius [34]; then, the P-E hysteresis loops of all samples become long and narrow, indicating that the relaxation property is improved, ferroelectricity is reduced and high energy storage efficiency will be possessed [33]. For each sample, the increase in E makes *P_max_* scale up and *P_r_* fluctuate slightly, stemming from the greater polarization response to the larger electric field [38]. According to Equation (1), the *W_rec_* value will increase accordingly. From Figure 9a, *P_max_* under 50 kV/cm of BSBiTZ-0.025SLT sintered at 1125 °C and 1155 °C is obviously higher than the other sintering temperatures since smaller grains generate more space charges in the grain boundaries owing to the occurrence of point defects such as electrons and oxygen vacancies, resulting in space charge polarization and, in turn, enhancing *P_max_* [15]. The *P_max_* value at sintering temperature 1155 °C is 3.02 μC/cm^2^, which outweighs 2.54 μC/cm^2^ for the 1140 °C sintered sample and can be decided by the morphology uniformity, excellent crystallinity, less porosity and highest density. When the sintering temperature exceeds 1170 °C, porosity increases again and abnormal grain growth appears, leading to the decrease in *P_max_*, which is consistent with the change of dielectric constant. To further reveal the effect of polarization on the energy-storage properties, the P-E hysteresis loops of BSBiTZ-xSLT at the same electric field of 50 kV/cm are shown in Figure 9b. The corresponding values of *P_max_*, P_r_, *W_t_* (total energy storage density), *W_rec_* and *η* at 50 kV/cm are given in Appendix A. The values of *P_max_*, *W_rec_* and *W_t_* gradually decrease with an increase in the x value, which is related to the increase in grain size and the random electric field induced by adding SLT [15]. The *η* value increases slightly first and then decreases gradually, but is still rather high. From Equations (2)–(5), it can be seen that point defects with opposite charges can form defect dipoles, which will be bound by adversely charged defects to reduce leakage current and increase breakdown field strength, inhibiting grain growth [15,24]. The decreased *P_max_* value of the BSBiTZ-0.075SLT and BSBiTZ-0.1SLT ceramics is relevant to the decrease in dielectric constant originating from the larger grain size and increased heterogeneity. The variation of the *P_r_* value is similar to that of *P_max_*. The slow increase in the *P_r_* value indicates the transition from partial polar nanodomains to ferroelectric domains [33,36]. Although the *P_r_* value and the *P_max_* value present a similar change trend, the increase rate of P_max_ is much faster than P_r_ due to the relaxation characteristic and phase structure [11,25]. Hence, the *W_rec_* and *η* values reach a crest at x = 0.025, at 67.61 mJ/cm^3^ and 97.02%, respectively, due to the improved relaxor feature and the larger *ΔP* difference between *P_max_* and *P_r_*.

The pulse charge–discharge performance is extremely important for the practical application of pulse energy storage capacitors. Figure 10 and Appendix A show the overdamped discharge current, discharge density and underdamped discharge current curves diagrams of the BSBiTZ-xSLT ceramics under 300 Ω load and the external electric field from 10 kV/cm to 30 kV/cm. From the overdamped discharge current curves, it can be seen that the current grows steadily after the discharge time of 450 ns under all the electric fields. The discharge time corresponding to the 90% discharge energy density (t_0.9_) is a significant parameter for the pulse charge–discharge properties, which is marked on the discharge density curves. The t_0.9_ value shifts to shorter discharge time with increasing the electric field and all t_0.9_ time is less than 160 ns under different electric fields, displaying extremely fast discharging performance. The variation of *W_d_* with the increase in the external electric field presents a similar change tendency. As we know, all the dipole moments gradually turn to the electric field and the polarization becomes saturation during the charge process. Then, the switched dipole moments suddenly go back to the initial state and release energy [38]. On the other hand, the larger electric field and the smaller dipoles are helpful to the high-speed conversion of dipoles. Thus, the above larger *W_d_* and the smaller t_0.9_ indicate that the long-range ordered ferroelectric domains are successfully converted to the small-size PNRs due to heterovalent ions doping and adding SLT. From all the underdamped discharge current diagrams, it can be seen that the discharge current reaches the maximum value at around 30 ns and decays to 0 after two oscillations. Such curves show the occurrence of peaks with an increasing electric field, which is related to the strength and degeneration of internal resistance. The enhanced driving force obtained from the increased electric field will make the carriers readily move [38]. To further understand the influence of sintering temperature and composition on the discharge mechanism, the discharge current and discharge density of different compositions at different sintering temperatures under the same electric field of 30 kV/cm are displayed in Appendix A. With the increase in the sintering temperature, the discharge density value of the BSBiTZ-0.025SLT ceramics decreases first and finally increases slightly, which is related to the variation of grain size and dielectric constant. The substantial increase in grain size with elevating the sintering temperature brings the increased conductivity and the rebound of dielectric constant at the sintering temperature of 1185 °C, which leads to a slight increase in the discharge density and discharge current [39,40]. Meanwhile, the discharge density and discharge current decrease with the increase in SLT doping amount since the PNR quantity gradually decreases with the addition of SLT and the increase in grain size [34,37]. The 1155 °C sintered BSBiTZ-0.025SLT ceramics possess the discharge density, discharge current, current density and power density of 32.13 mJ/cm^3^, 11.16 A, 1016.71 A/cm^2^ and 305.01 MW/cm^3^ under 30 kV/cm, respectively, showing better pulse charge–discharge characteristics and presenting a prospective application in the field of pulse energy storage.

## 4. Conclusions

In this work, the (1-x)(Ba_0.75_Sr_0.1_Bi_0.1_)(Ti_0.9_Zr_0.1_)O_3_-x(Sb_0.5_Li_0.5_)TiO_3_ (BSBiTZ-xSLT, x = 0.025, 0.05, 0.075, 0.1) ceramics were prepared by a conventional solid-state sintering method, whose predominant phase is the perovskite structure. All ceramics have phase coexistence of tetragonal, rhombohedral and orthogonal phases, confirmed by the XRD Rietveld refinement and Raman spectroscopy analysis. The optimal sintering temperature for preparing the BSBiTZ-0.025SLT, BSBiTZ-0.05SLT, BSBiTZ-0.075SLT and BSBiTZ-0.1SLT ceramics is 1155 °C, 1155 °C, 1155 °C and 1170 °C, respectively, which have highest relative density, least gas pores, densified micro-morphology and mainly spherical-like shaped grains due to a liquid-phase sintering mechanism. Obvious abnormal grain growth appears and grain size distribution departs from the normal distribution when the sintering temperature is above 1155 °C and the SLT amount is high. All BSBiTZ-xSLT ceramics are tailored to present relaxor ferroelectric characteristics, the dielectric constant plateau is expanded to ~120 °C with increased temperature stability with variation less than 10% and very long and narrow hysteresis loops are acquired. The 1155 °C sintered BSBiTZ-0.025SLT ceramics possess the maximum energy storage density and energy storage efficiency, at 65.59 mJ/cm^3^ and 97.02% under 50 kV/cm, respectively, correlating with the highest relative density (96.60%) and least porosity, maximum dielectric constant, excellent crystallinity and increased morphology uniformity with fine grain size. The t_0.9_ value shifts to a shorter discharge time with increasing the electric field and all t_0.9_ time is less than 160 ns under different electric fields, displaying extremely fast discharge performance. The BSBiTZ-0.025SLT ceramics reach the largest discharge current at around 30 ns and the attenuation is 0 after two oscillations, showing excellent pulse charge–discharge performance and presenting remarkable application value in the field of pulse energy storage.

## Figures and Tables

**Figure 1 materials-17-02085-f001:**
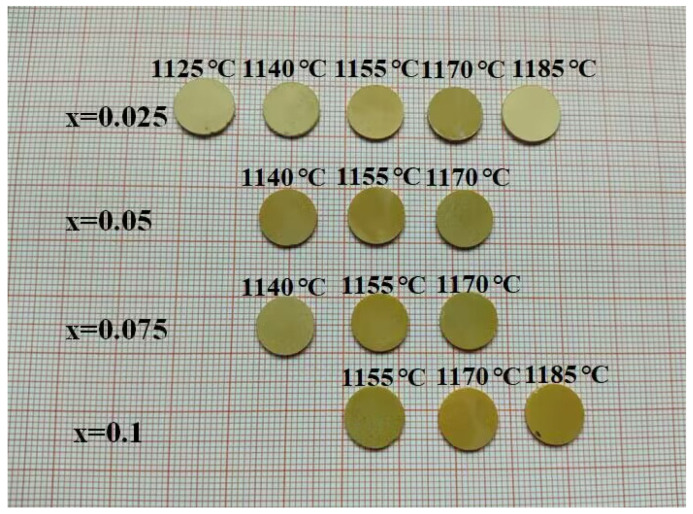
The sintered BSBiTZ-xSLT (x = 0.025, 0.05, 0.075, 0.1) ceramics prepared at different sintering temperatures.

**Figure 2 materials-17-02085-f002:**
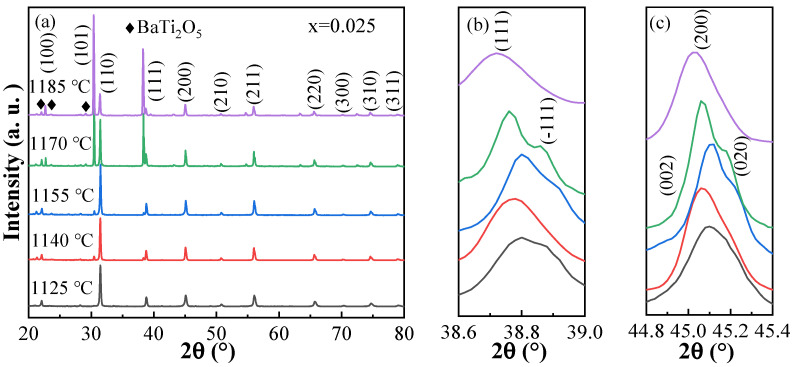
(**a**) XRD patterns of BSBiTZ-0.025SLT ceramics prepared at different sintering temperatures; (**b**) the enlarged XRD views of {111} diffraction peak; (**c**) the enlarged XRD views of {200} diffraction peak.

**Figure 3 materials-17-02085-f003:**
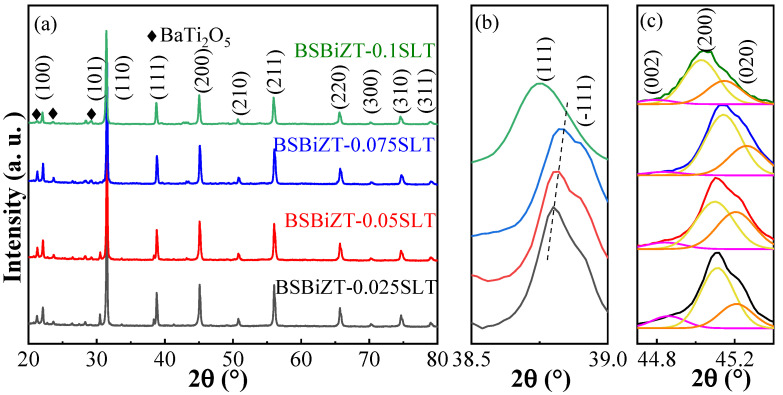
(**a**) XRD plots of BSBiTZ-xSLT (x = 0.025, 0.05, 0.075, 0.1) ceramics; (**b**) the enlarged XRD views of {111} diffraction peak; (**c**) the enlarged XRD views of {200} diffraction peak.

**Figure 4 materials-17-02085-f004:**
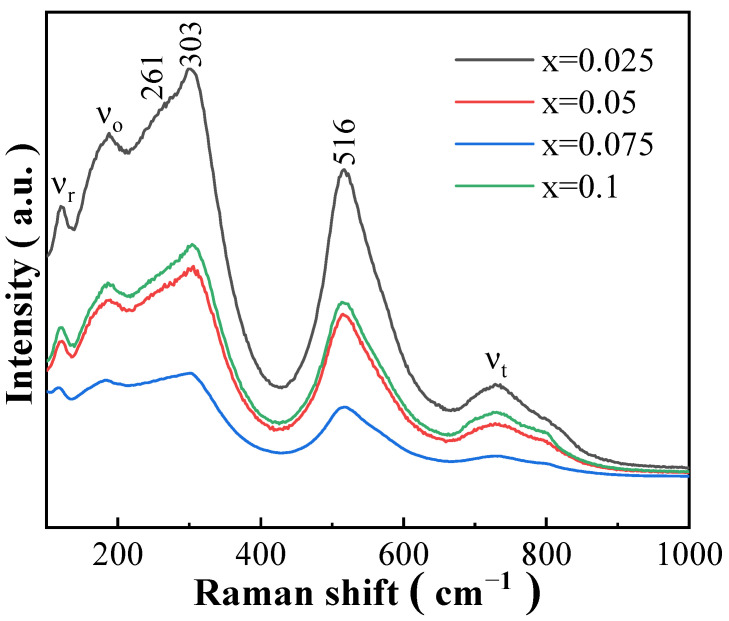
Raman spectra of BSBiTZ-xSLT (x = 0.025, 0.05, 0.075, 0.1) ceramics.

**Figure 5 materials-17-02085-f005:**
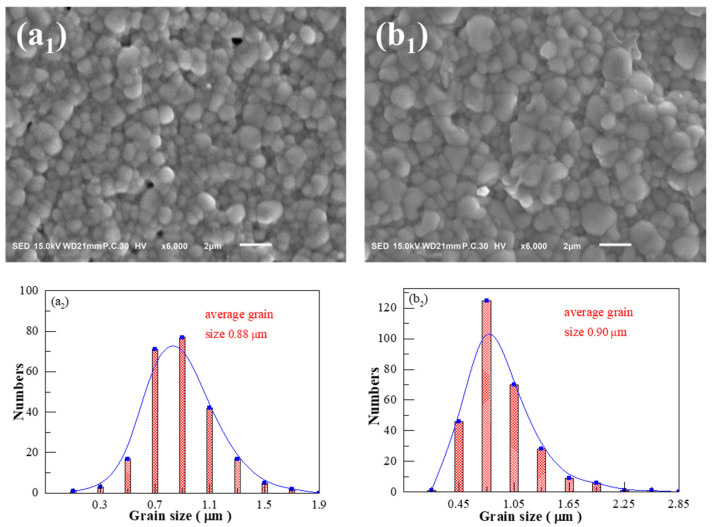
SEM photographs and grain size statistics of BSBiTZ-0.025SLT ceramics sintered at (**a1**) 1125 °C; (**b1**) 1140 °C; (**c1**) 1155 °C; (**d1**) 1170 °C; (**e1**) 1185 °C; (**a2**–**e2**) grain size distribution histogram curve.

**Figure 6 materials-17-02085-f006:**
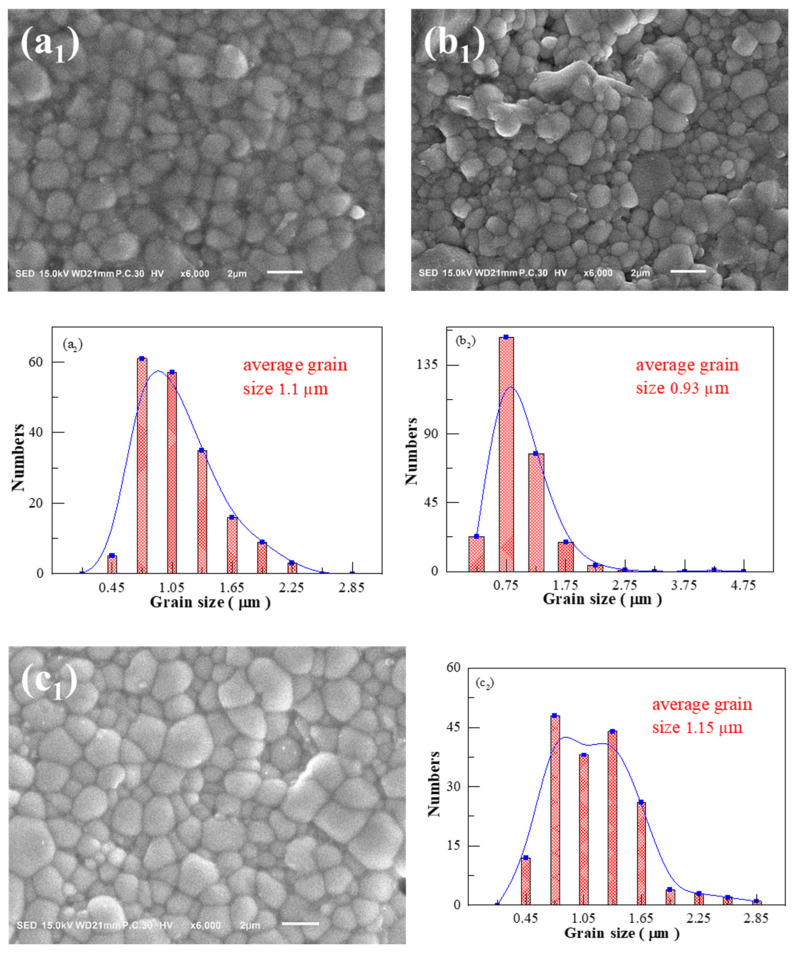
SEM images and corresponding grain size statistics curves. (**a1**) x = 0.05; (**b1**) x = 0.075; (**c1**) x = 0.1; (**a2**–**c2**) grain size distribution histogram curve.

**Figure 7 materials-17-02085-f007:**
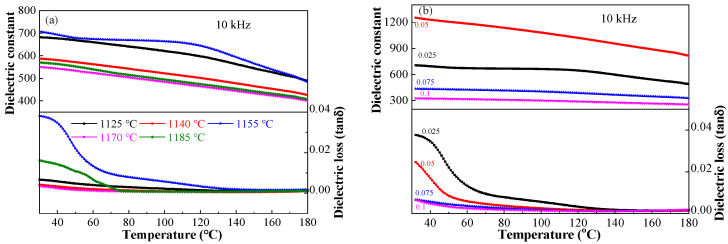
(**a**) Dielectric–temperature curves of BSBiTZ-0.025SLT ceramic at different sintering temperatures measured at 10 kHz; (**b**) dielectric–temperature curves of BSBiTZ-xSLT (x = 0.025, 0.05, 0.075, 0.1) ceramics prepared at respective optimal sintering temperature measured at 10 kHz.

**Figure 8 materials-17-02085-f008:**
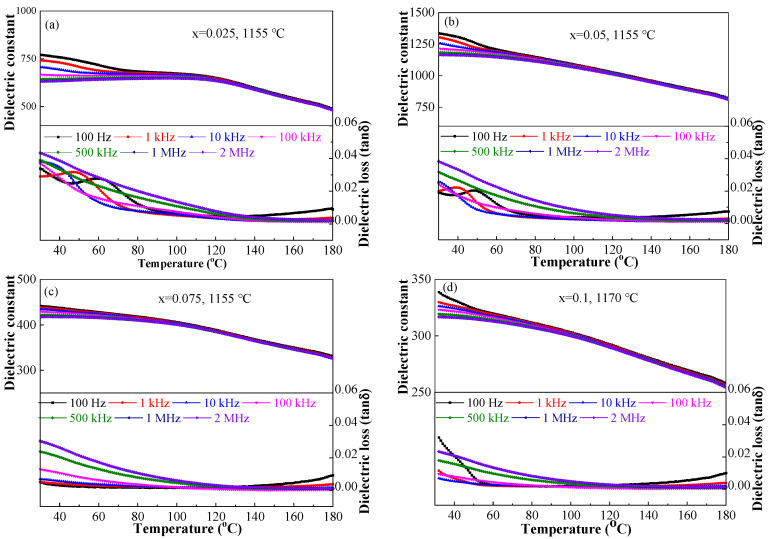
Dielectric–temperature curves of BSBiTZ-xSLT ceramics at the optimal sintering temperature measured at different frequencies. (**a**) x = 0.025; (**b**) x = 0.05; (**c**) x = 0.075; (**d**) x = 0.1.

**Figure 9 materials-17-02085-f009:**
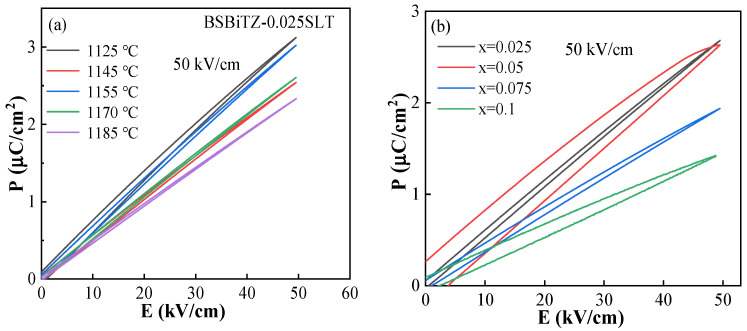
(**a**) Unipolar P-E hysteresis loops of BSBiTZ-0.025SLT ceramics at different sintering temperatures at 50 kV/cm; (**b**) unipolar P-E hysteresis loops of BSBiTZ-xSLT (x = 0.025, 0.05, 0.075, 0.1) ceramics at optimal sintering temperature at 50 kV/cm.

**Figure 10 materials-17-02085-f010:**
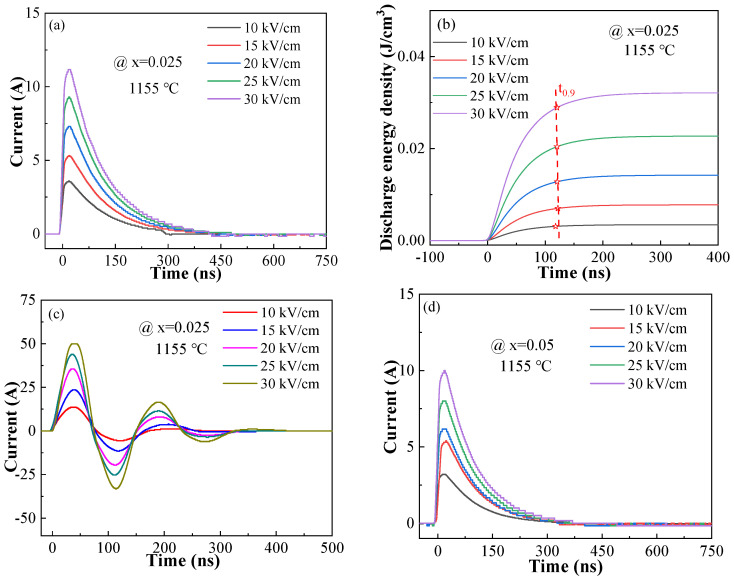
The pulse charge–discharge energy storage properties of BSBiTZ-xSLT. (**a**) Overdamped discharged voltage curves of BSBiTZ-0.025SLT; (**b**) the variation of discharge energy density and t_0.9_ with time of BSBiTZ-0.025SLT at 10–30 kV/cm; (**c**) underdamped discharged voltage curves of BSBiTZ-0.025SLT; (**d**) overdamped discharged voltage curves of BSBiTZ-0.05SLT; (**e**) the variation of discharge energy density and t_0.9_ with time of BSBiTZ-0.05SLT at 10–30 kV/cm; (**f**) underdamped discharged voltage curves of BSBiTZ-0.05SLT; (**g**) overdamped discharged voltage curves of BSBiTZ-0.075SLT; (**h**) the variation of discharge energy density and t_0.9_ with time of BSBiTZ-0.075SLT at 10–30 kV/cm; (**i**) underdamped discharged voltage curves of BSBiTZ-0.075SLT; (**j**) overdamped discharged voltage curves of BSBiTZ-0.1SLT; (**k**) the variation of discharge energy density and t_0.9_ with time of BSBiTZ-0.1SLT at 10–30 kV/cm; (**l**) underdamped discharged voltage curves of BSBiTZ-0.1SLT.

**Table 1 materials-17-02085-t001:** Bulk density and theoretical density of BSBiTZ-xSLT (x = 0.025, 0.05, 0.075, 0.1) ceramics at different sintering temperatures.

	Sintering Temperature (°C)	Bulk Density (g/cm^3^)	Theoretical Density (g/cm^3^)	Relative Density (%)
	1125	5.4254	5.9216	91.62
	1140	5.4616	5.8723	93.01
x = 0.025	1155	5.7180	5.9189	96.60
	1170	5.6355	5.9280	95.06
	1185	5.5578	5.9033	94.19
	1140	5.4616	5.8787	92.90
x = 0.05	1155	5.7366	5.8714	97.70
	1170	5.4239	5.8642	92.49
	1140	5.0021	5.8583	85.38
x = 0.075	1155	5.6904	5.8366	97.50
	1170	5.2314	5.8276	89.77
	1155	5.5432	5.7544	96.33
x = 0.1	1170	5.7188	5.7597	99.29
	1185	5.5757	5.7731	96.58

## Data Availability

All data that support the findings of this study are included within the article and Appendix A.

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
