# Peer review of "(Sb0.5Li0.5)TiO3-Doping Effect and Sintering Condition Tailoring in BaTiO3-Based Ceramics"

_materials, 2024, doi:10.3390/ma17092085_

Round 1

Reviewer 1 Report

Comments and Suggestions for Authors

I read manuscripts on ceramic preparation. However, some issues remain unclear. This needs to be resolved before further consideration is given to whether to accept publication.

1. The abstract is too long and makes no real difference in results. For example, temperature changes will vary several times, etc.

2. The hypothesis of this study should be stated in the Introduction.

3. The purpose of this study is not clearly stated in the Introduction.

4. “The 1155 ℃ sintered 0.975(Ba0.75Sr0.1Bi0.1)(Ti0.9Zr0.1)O3-0.025(Sb0.5Li0.5)TiO3 ceramics show good temperature stability of dielectric constant till ~120 °C with variation less than 10%, excellent energy storage performance with Wrec=65.59 mJ/cm3 and η=97.02% under 50 kV/cm, and the discharge current (Id) reaches the maximum value at around 30 ns, providing remarkable application value in the fields of pulse power supply and electric vehicle [23].”.

This sentence is very strange. Is it a report describing previous research?

5. Experimental Procedure is too simple. The experimental groups, steps and processes should be explained in detail.

6. There is a lack of statistical analysis methods in the materials methods.

7. 1185 degrees, 1170 degrees, 1155 degrees, 1140 degrees and 1125 degrees. What are these conditions based on?

8. Each group in Table 1. should show the same temperature. However, some groups are missing. In addition, there is no standard deviation and statistical analysis.

9. Figure 4. Shows the grain size statistics of BSBiTZ-0.025SLT at different temperatures. However, Figure 5. shows the results at different concentrations and a single temperature. This is wrong, the author cannot just present the results under different lines. It should be possible to show differences between different groups under the same conditions.

Likewise, the results are the same in other cases.

10. The manuscript lacks quantitative analysis and is almost entirely qualitative. This fails to explain the difference and is unscientific. Because there is no statistical analysis.

11. Conclusions are not discussions. Must be rewritten.

6.

Comments on the Quality of English Language

 Moderate editing of English language required

Author Response

  1. The abstract is too long and makes no real difference in results. For example, temperature changes will vary several times, etc.

Answer:

Thank you for your suggestion. The abstract was revised to be more concise and rarely identical to the results.

Abstract: (1-x)(Ba0.75Sr0.1Bi0.1)(Ti0.9Zr0.1)O3-x(Sb0.5Li0.5)TiO3 (abbreviated as BSBiTZ-xSLT, x=0.025, 0.05, 0.075, 0.1) ceramics were prepared via conventional solid-state sintering method under different sintering temperatures. All BSBiTZ-xSLT ceramics have predominantly perovskite phase structure with the coexistence of tetragonal, rhombohedral and orthogonal phases, and present mainly spherical-like shape grains relating to liquid-phase sintering mechanism due to adding SLT and Bi2O3. By adjusting sintering temperature, all compositions obtain highest relative density and present densified micro-morphology, and doping SLT tends to promote the growth of grain size and the grain size distribution becomes nonuniform gradually. Due to adding heterovalent ions and SLT, typical relaxor ferroelectric characteristic is realized, dielectric performance stability is broadened to ~120 °C with variation less than 10%, and very long and slim hysteresis loops are obtained, which is especially beneficial for energy storage application. All samples show extremely fast discharge performance where the discharge time t0.9 (time for 90% discharge energy density) is less than 160 ns and the largest discharge current occurs at around 30 ns. The 1155 °C sintered BSBiTZ-0.025SLT ceramics exhibit rather large energy storage density, very high energy storage efficiency and excellent pulse charge-discharge performance, providing possibility to develop novel BT-based dielectric ceramics for pulse energy storage application.”

  1. The hypothesis of this study should be stated in the Introduction.

Answer:

The hypothesis of the study was added in the introduction.

“Therefore, adding heterovalent ions Sb3+ and Li+ and second component SLT into the BT-based ceramics was undertaken in this work to realize relaxation characteristic, where Li improves relaxation performance due to inequality of electric valence and ionic radius, Sb balances the electrovalence caused by Li doping, strengthens the tetragonal phase and increases microscopic morphology homogeneity, and SLT not only improves the relaxation properties but also reduces the sintering temperature.”

“Therefore, the study aims to obtain high density, small grain size and uniform microscopic morphology can be obtained by adjusting the sintering temperature. At the same time, the TC temperature can be adjusted below the room temperature and the resistance of ceramics can be improved combined with composition designing.”

  1. The purpose of this study is not clearly stated in the Introduction.

Answer:

The purpose of the study was added in the introduction.

“In order to develop novel green and environment friendly lead-free dielectric ceramics for pulse energy storage application, the BT-based ceramics were studied with adding heterovalent ions Sb3+ and Li+ and second component SLT. The effects of sintering temperature and (Sb0.5Li0.5)TiO3 addition amount on structure, morphology and electrical properties of (Ba0.75Sr0.1Bi0.1)(Ti0.9Zr0.1)O3 were researched systematically, and basic mechanism of enhanced thermal stability was revealed.”

  1. “The 1155 ℃ sintered 0.975(Ba0.75Sr0.1Bi0.1)(Ti0.9Zr0.1)O3-0.025(Sb0.5Li0.5)TiO3 ceramics show good temperature stability of dielectric constant till ~120 °C with variation less than 10%, excellent energy storage performance with Wrec=65.59 mJ/cm3 and η=97.02% under 50 kV/cm, and the discharge current (Id) reaches the maximum value at around 30 ns, providing remarkable application value in the fields of pulse power supply and electric vehicle [23].”.

This sentence is very strange. Is it a report describing previous research?

Answer:

The sentence described the major conclusion and highlight of this work, which was revised as below.

“Eventually, in this work densified sintering with relative density exceeded 95% is realized in the (1-x)(Ba0.75Sr0.1Bi0.1)(Ti0.9Zr0.1)O3-x(Sb0.5Li0.5)TiO3 system, and the 1155 ℃ sintered 0.975(Ba0.75Sr0.1Bi0.1)(Ti0.9Zr0.1)O3-0.025(Sb0.5Li0.5)TiO3 ceramics have excellent energy storage performance, which show good temperature stability of dielectric constant till ~120 °C with variation less than 10%, excellent energy storage performance with Wrec=65.59 mJ/cm3 and η=97.02% under 50 kV/cm, and the discharge current (Id) reaches the maximum value at around 30 ns, providing remarkable application value in the fields of pulse power supply and electric vehicle [23].”

  1. Experimental Procedure is too simple. The experimental groups, steps and processes should be explained in detail.

Answer:

The experimental procedure was revised as below.

“Bulk (1-x)(Ba0.75Sr0.1Bi0.1)(Ti0.9Zr0.1)O3-x(Sb0.5Li0.5)TiO3 (abbreviated as BSBiTZ-xSLT, x=0.025, 0.05, 0.075, 0.1) ceramics were prepared by conventional solid-state reaction method. The raw reagents, BaCO3 (Sinopharm Chemical Reagent Co., Ltd, China, Shanghai), SrCO3 (Sinopharm Chemical Reagent Co., Ltd, China, Shanghai), Bi2O3 (Sinopharm Chemical Reagent Co., Ltd, China, Shanghai), TiO2 (Sinopharm Chemical Reagent Co., Ltd, China, Shanghai), ZrO2 (Sinopharm Chemical Reagent Co., Ltd, China, Shanghai), Sb2O3 (Shanghai Shisihewei Chemical Co., Ltd, China, Shanghai) and Li2CO3 (Shanghai Shanhai Gongxue Group Experiment No.2 Factory, China, Shanghai), were stoichiometrically weighed after full drying, fully ground and mixed, and passed through a 100-mesh sieve. The sieved powder was calcined in a muffle furnace whose temperature was increased to 500 °C for 167 min, then raised to 925 °C at 5 °C/min, and finally kept at 925 °C for 3 hours. Polyvinyl alcohol (PVA) solution was mixed with the calcined powder for granulation, and the granulated powder was passed through a 80-mesh sieve. The granulated powder was pressed into green pellets with diameter of 10 mm and thickness of about 1 mm via cold-pressing under a pressure of 350 MPa and holding time of 1 min. Subsequently, the pressed discs were placed in a muffle furnace and increased to 550 °C by 200 min and incubated for 2 h to remove the PVA. Then, using ZrO2 as the covering powder to sinter the decarburized pressed discs at different temperatures with holding time 2 hours and the sintered ceramics are shown in Figure 1. Several sintering temperatures were treated with 15 °C as interval depending on composition, and sintering temperature range and optimized sintering temperature were determined based on density measurement and performance characterization.”

  1. There is a lack of statistical analysis methods in the materials methods.

Answer:

For physical performance characterization, statistical method was seldom used.

The grain size distribution was obtained by a statistical analysis method. The grain size was measured by a linear intersection method using the Nano Measure software, based on which grain size distribution of ceramics was obtained and discussed in the micromorphology analysis section.

“SEM photos and corresponding grain size distribution statistical curves of the BSBiTZ-0.025SLT ceramics measured by a linear intersection method using the Nano Measure software sintered at 1125 ℃, 1140 ℃, 1155 ℃, 1170 ℃ and 1185 ℃ are shown in Figure 5.”

“The grain size increases gradually from 0.88 μm at 1125 ℃ to 1.52 μm at 1185 ℃, and grain size distribution departs from the normal distribution accompanied by obvious abnormal grain growth above sintering temperature 1155 ℃……”

  1. 1185 degrees, 1170 degrees, 1155 degrees, 1140 degrees and 1125 degrees. What are these conditions based on?

Answer:

Comment 9 also discussed the selection of sintering temperature.

Sintering condition was important parameter to enhance electrical performance. The sintering temperature of (Ba1-xLix)TiO3 given in Ref. [12] was 1200 °C, which was treated as the first sintering temperature point in this study and 15 °C was used as intervals to search sintering temperature range and optimized sintering temperature. The relative density and electrical performance were used to judge optimized sintering temperature.

The adjusting the sintering temperature was added to the experiment procedure.

  1. Each group in Table 1. should show the same temperature. However, some groups are missing. In addition, there is no standard deviation and statistical analysis.

Answer:

Please combine with answering for comment 9 for this comment.

In this study, the structure and properties were studied by two aspects, composition and sintering temperature.

First, the BSBiTZ-0.025SLT ceramics were studied systematically. Based on the relationship between the relative density and sintering temperature, sintering temperature range and optimized sintering temperature were determined. Then, SEM micro-morphology could confirm such judgement, and grain size distribution was obtained.

Then, the sintering temperature range and the optimal sintering temperature of the other components were obtained based on the above obtained optimized sintering temperature. The optimal sintering temperature of the other compositions could be obtained by decreased number of sintering experiments.

For x=0.1 composition, slightly higher sintering temperature was required to acquire sintering temperature range based on the relative density measurement.

  1. Figure 4. Shows the grain size statistics of BSBiTZ-0.025SLT at different temperatures. However, Figure 5. shows the results at different concentrations and a single temperature. This is wrong, the author cannot just present the results under different lines. It should be possible to show differences between different groups under the same conditions.

Likewise, the results are the same in other cases.

Answer:

This work was to develop energy storage application ceramics, where the influence of second component (Sb0.5Li0.5)TiO3 and sintering temperature were studied.

x=0.025 composition was explored systematically first, based on which other compositions were studied with decreased number of sintering experiments. For x=0.1 composition, slightly higher sintering temperature was required to acquire sintering temperature range based on the relative density measurement.

According to this thinking, in the micro-morphology section and grain size distribution analysis, i.e., Figure 4 and Figure 5, SEM studied micro-morphology and compared with density, and the change of grain size with the increase of the sintering temperature and SLT amount was studied. These factors determined energy storage performance mainly.

First, the BSBiTZ-0.025SLT ceramics were studied the relationship between grain size and sintering temperature, showing that the grain size increased with the sintering temperature increasing. Such conclusion was correct for all compositions.

Then, the influence of SLT content was studied for samples under the optimal sintering temperature. With the increase of SLT amount, the grain size increases first and then decreases, and finally increases, and the micro-morphology changes from uniform to nonuniform.

Densified morphology, small grain size and uniform micro-morphology will increase breakdown field strength. Excessive grain growth reduces the density of grain boundary and decreases the breakdown field strength, which deteriorates energy storage performance.

  1. The manuscript lacks quantitative analysis and is almost entirely qualitative. This fails to explain the difference and is unscientific. Because there is no statistical analysis.

Answer:

For physical performance characterization, statistical method was seldom used, which was same to the cited references.

The grain size distribution was obtained by a statistical analysis method and the quantitative analysis was added in the manuscript.

  1. Conclusions are not discussions. Must be rewritten.

Answer:

The conclusions were rewritten.

“In this work, the (1-x)(Ba0.75Sr0.1Bi0.1)(Ti0.9Zr0.1)O3-x(Sb0.5Li0.5)TiO3 (BSBiTZ-xSLT, x=0.025, 0.05, 0.075, 0.1) ceramics were prepared by conventional solid-state sintering method, whose predominantly phase is perovskite structure. All ceramics have phase coexistence of tetragonal, rhombohedral and orthogonal phases confirmed by the XRD Rietveld refinement and Raman spectroscopy analysis. The optimal sintering temperature for preparing the BSBiTZ-0.025SLT, BSBiTZ-0.05SLT, BSBiTZ-0.075SLT and BSBiTZ-0.1SLT ceramics is 1155 °C, 1155 °C, 1155 °C and 1170 °C, respectively, which have highest relative density, least gas pores, densified micro-morphology and mainly spherical-like shape grains due to liquid-phase sintering mechanism. Obvious abnormal grain growth appears and grain size distribution departs from the normal distribution when sintering temperature is above 1155 ℃ and the SLT amount is high. All BSBiTZ-xSLT ceramics are tailored to present relaxor ferroelectric characteristic, the dielectric constant plateau is expanded to ~120 °C with increased temperature stability with variation less than 10% and very long and narrow hysteresis loops are acquired. The 1155 °C sintered BSBiTZ-0.025SLT ceramics possess the maximum energy storage density and energy storage efficiency, being 65.59 mJ/cm3 and 97.02% under 50 kV/cm, respectively, correlating with the highest relative density (96.60%) and least porosity, maximum dielectric constant, excellent crystallinity, and increased morphology uniformity with fine grain size. The t0.9 value shifts to shorter discharge time with increasing the electric field and all t0.9 time is less than 160 ns under different electric fields, displaying extremely fast discharge performance. The BSBiTZ-0.025SLT ceramics reach the largest discharge current at around 30 ns and the attenuation is 0 after two oscillations, showing excellent pulse charge-discharge performance and presenting remarkable application value in the field of pulse energy storage.”

Reviewer 2 Report

Comments and Suggestions for Authors

Major issue: there were used 7 reagents (a large number), some of them being added in very small amounts as compared to others. Please provide details, in the manuscript, of how did you managed to obtain an uniform distribution of the oxides as the reagents were mixed (via grinding, as mentioned in the manuscript). In this case, poor mixing could be expected and, consequently, poor chemical uniformity.

Minor issues:

- line 220, to correct the text on the abscissae;

- line 276, it is Li2CO3 or Li2O?

Author Response

Major issue: there were used 7 reagents (a large number), some of them being added in very small amounts as compared to others. Please provide details, in the manuscript, of how did you managed to obtain an uniform distribution of the oxides as the reagents were mixed (via grinding, as mentioned in the manuscript). In this case, poor mixing could be expected and, consequently, poor chemical uniformity.

Answer:

More reagents were used to tailor BaTiO3 into relaxor characteristic, which could enhance energy storage performance. Such method was widely used in related research.

As the reviewer said, chemical uniformity was very important in such research. And the mixing was not a problem in modern technique. Ball milling, grinding, and mixing were done repeatedly to increase the mixing uniformity, and EDX mapping could confirm the relatively uniform distribution of different elements.

Minor issues:

- line 220, to correct the text on the abscissae;

Answer:

Thank you for your reminder and the abscissae was revised.

- line 276, it is Li2CO3 or Li2O?

Answer:

It is Li2CO3, which was decomposed into Li2O during calcining and solid soluted into the perovskite structure during sintering.

Reviewer 3 Report

Comments and Suggestions for Authors

The paper “(Sb0.5Li0.5) Ti O3 doping effect and sintering condition tailoring in BaTiO3-based ceramics ” presents samples of a complex composition, BaTiO3 doped with Sb and Li ions and sintered at different temperatures. I find this study supple, full and extensive. I found only minor gaps that can be easily filled upon minor revision.

1. Line 84. Authors mention the value η, but do not decipher what it means.

2. Why did you choose exactly these x = 0.025, 0.05, 0.075 and 0.1? Why with the step of 0.025?

3. Experimental Section

All mentioned equipment should be presented in the form Model (Manufacturer, country, city).

Mentioned chemical should be presented in the form ABO3 (Manufacturer, country, city).

 If you have a photograph of samples after sintering, could you provide them in this paper? I find it useful in case if someone would like to reproduce your experiment.

4. Figure 1: I recommend separating the denotation of temperatures somewhere near the graphs. Picture is too jammed.

Figure 2: the same.

5. Ref [27] – is it correct to cite the paper describing phases in (Na,K) NbO3 in your case?

6. Wavenumber is usually denoted in Raman spectroscopy as Greek letter ν, not Latin V.

7. Figure 3: Raman shift, not Roman.

8. WinPLOT fitting and Jade refinement should be described in greater detail in the Methods section. At least authors should give a reference to the methods description.

9. Table 1: Could you please separate compositions with different x by a horizontal line?

Comments on the Quality of English Language

10 English

Increase/decrease is always IN.

Line 188: AS high as…

Author Response

The paper “(Sb0.5Li0.5)TiO3 doping effect and sintering condition tailoring in BaTiO3-based ceramics ” presents samples of a complex composition, BaTiO3 doped with Sb and Li ions and sintered at different temperatures. I find this study supple, full and extensive. I found only minor gaps that can be easily filled upon minor revision.

  1. Line 84. Authors mention the value η, but do not decipher what it means.

Answer:

Æž was energy storage efficiency. The meaning of η was       .

  1. Why did you choose exactly these x = 0.025, 0.05, 0.075 and 0.1? Why with the step of 0.025?

Answer:

This work was to develop energy storage application ceramics, which was realized by doping metals and adding second component into the BaTiO3 matrix.

(Sb0.5Li0.5)TiO3 was chosen as the second component, and x=0.025 was used as step to search the appropriate content to obtain excellent relaxor characteristic, which was necessary to enhance energy storage performance. Moreover, in the previous reference, 0.02, 0.25 or 0.05 were used as step to study appropriate composition depending on ferroelectric system.

Sintering condition was also important parameter to enhance electrical performance. In this study, the first component x=0.025 was explored systematically, and sintering temperature range and optimized sintering temperature were determined. Based on which compositions x=0.05, x=0.075 and x=0.1 were studied, and optimized sintering temperature could be obtained with decreased number of sintering experiments. We could see the dielectric properties and electrical properties deteriorated with the increase of x.

x=0.1 composition required slightly higher sintering temperature to present density decrease trend. Besides, x=0.1 composition showed increased grain size significantly, the grain distribution was uneven, and the polarization strength and energy storage density were greatly reduced. So, composition with x>0.1 was not studied further.

  1. Experimental Section

All mentioned equipment should be presented in the form Model (Manufacturer, country, city).

Mentioned chemical should be presented in the form ABO3 (Manufacturer, country, city).

 If you have a photograph of samples after sintering, could you provide them in this paper? I find it useful in case if someone would like to reproduce your experiment.

Answer:

The equipment and chemical reagents were revised according to the pattern and the photograph of samples were added in Figure 1.

“Bulk (1-x)(Ba0.75Sr0.1Bi0.1)(Ti0.9Zr0.1)O3-x(Sb0.5Li0.5)TiO3 (abbreviated as BSBiTZ-xSLT, x=0.025, 0.05, 0.075, 0.1) ceramics were prepared by conventional solid-state reaction method. The raw reagents, BaCO3 (Sinopharm Chemical Reagent Co., Ltd, China, Shanghai), SrCO3 (Sinopharm Chemical Reagent Co., Ltd, China, Shanghai), Bi2O3 (Sinopharm Chemical Reagent Co., Ltd, China, Shanghai), TiO2 (Sinopharm Chemical Reagent Co., Ltd, China, Shanghai), ZrO2 (Sinopharm Chemical Reagent Co., Ltd, China, Shanghai), Sb2O3 (Shanghai Shisihewei Chemical Co., Ltd, China, Shanghai) and Li2CO3 (Shanghai Shanhai Gongxue Group Experiment No.2 Factory, China, Shanghai), were stoichiometrically weighed after full drying, fully ground and mixed, and passed through a 100-mesh sieve. The sieved powder was calcined in a muffle furnace whose temperature was increased to 500 °C for 167 min, then raised to 925 °C at 5 °C/min, and finally kept at 925 °C for 3 hours. Polyvinyl alcohol (PVA) solution was mixed with the calcined powder for granulation, and the granulated powder was passed through a 80-mesh sieve. The granulated powder was pressed into green pellets with diameter of 10 mm and thickness of about 1 mm via cold-pressing under a pressure of 350 MPa and holding time of 1 min. Subsequently, the pressed discs were placed in a muffle furnace and increased to 550 °C by 200 min and incubated for 2 h to remove the PVA. Then, using ZrO2 as the covering powder to sinter the decarburized pressed discs at different temperatures with holding time 2 hours and the sintered ceramics are shown in Figure 1. Several sintering temperatures were treated with 15 °C as interval depending on composition, and sintering temperature range and optimized sintering temperature were determined based on density measurement and performance characterization.

Crystal structure of the BSBiTZ-xSLT ceramics was determined by X-ray diffraction measurement (XRD, Rigaku D/max-2500/PC, Rigaku Corp., Tokyo, Japan). The peak splitting and accurate phase structure were analyzed by the WINPLOTR software. The theoretical density and the composition induced phase transformation were acquired by the MDI Jade 6.5 software and Raman spectroscopy (Horiba Scientific, Ltd., Kyoto, Japan) using a laser with a wavelength of 633 nm in the range of 100 ~ 1000 cm-1. Scanning electron microscopy (SEM, JEOL Ltd., Tokyo, Japan) and Nano Measure software were used to obtain the microstructure and grain size distribution, respectively. Dielectric properties were tested from room temperature to 180 °C at 100 Hz to 2 MHz using a Partulab HDMS-1000 measurement system (Partulab Technology Co. Ltd, Wuhan, China) combined with a Microtest Precision LCR Meter 6630-10 (Microtest Corp., Taiwan, China). The ferroelectric properties were tested at 10 Hz using a ferroelectric analyzer (Radiant Technologies Inc., Albuquerque, New Mexico, USA). The resistance-inductance-capacitance test system (Shanghai Tongguo Intelligent Technology Co., Ltd., Shanghai, China) was used to test out the pulse charge-discharge performance.

Figure 1. The sintered BSBiTZ-xSLT (x=0.025, 0.05, 0.075, 0.1) ceramics at different sintering temperatures.

  1. Figure 1: I recommend separating the denotation of temperatures somewhere near the graphs. Picture is too jammed.

Figure 2: the same.

Answer:

The pictures were adjusted, and the number was changed to Figure 2 and Figure 3 due to adding one new figure of Figure 1.

  1. Ref [27] – is it correct to cite the paper describing phases in (Na,K) NbO3 in your case?

Answer:

The section explaining the phase structure with ref [27] was removed.

  1. Wavenumber is usually denoted in Raman spectroscopy as Greek letter ν, not Latin V.

Answer:

Thank you for your reminder. The V was revised by ν.

  1. Figure 3: Raman shift, not Roman.

Answer:

The Figure 3 was revised. Roman was corrected by Raman shift.

  1. WinPLOT fitting and Jade refinement should be described in greater detail in the Methods section. At least authors should give a reference to the methods description.

Answer:

The WINPLOTR fitting and Jade refinement was described in the Methods section.

“The peak splitting and accurate phase structure were analyzed by the WINPLOTR software. The theoretical density and the composition induced phase transformation were acquired by the MDI Jade 6.5 software and Raman spectroscopy (Horiba Scientific, Ltd., Kyoto, Japan) using a laser with a wavelength of 633 nm in the range of 100 ~ 1000 cm-1.”

  1. Table 1: Could you please separate compositions with different x by a horizontal line?

Answer:

I have separated compositions with different x by a horizontal line.

  1. English

Increase/decrease is always IN.

Line 188: AS high as…

Answer:

English was polished trying our best.

Round 2

Reviewer 1 Report

Comments and Suggestions for Authors

The manuscript has been appropriately revised. I think it's acceptable. Thanks.

Comments on the Quality of English Language

Minor editing of English language required

Reviewer 2 Report

Comments and Suggestions for Authors

I find the response satisfactory, thank you!